

# Many bounded versions of undecidable problems are NP-hard

Andreas Klingler[*], Mirte van der Eyden[†], Sebastian Stengele,
Tobias Reinhart and Gemma De las Cuevas

Institute for Theoretical Physics, Technikerstr. 21a, A-6020 Innsbruck, Austria

[*] Andreas.Klingler@uibk.ac.at , [†] Mirte.van-der-Eyden@uibk.ac.at

## Abstract

Several physically inspired problems have been proven undecidable; examples are the spectral gap problem and the membership problem for quantum correlations. Most of these results rely on reductions from a handful of undecidable problems, such as the halting problem, the tiling problem, the Post correspondence problem or the matrix mortality problem. All these problems have a common property: they have an NP-hard bounded version. This work establishes a relation between undecidable unbounded problems and their bounded NP-hard versions. Specifically, we show that NP-hardness of a bounded version follows easily from the reduction of the unbounded problems. This leads to new and simpler proofs of the NP-hardness of bounded version of the Post correspondence problem, the matrix mortality problem, the positivity of matrix product operators, the reachability problem, the tiling problem, and the ground state energy problem. This work sheds light on the intractability of problems in theoretical physics and on the computational consequences of bounding a parameter.

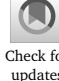

# 1   Introduction

Many problems in quantum information and quantum many-body physics are undecidable. This includes the spectral gap of physical systems [1, 2], membership problems for quantum correlations [3–7], properties of tensor networks [8–10], measurement occurrence and reachability problems [11, 12], and many more [13–17]. In addition, other problems are believed to be undecidable, such as detecting quantum capacity [18], distillability of entanglement [12], or tensor-stable positivity [14].

All these problems have a common theme: They ask for a property that includes an unbounded parameter. For example, in a quantum correlation scenario, the dimension of the shared quantum state between the two parties may be unbounded. Also properties of many-body systems, like the spectral gap, are statements involving arbitrarily large system sizes.

On the other hand, many problems in science, engineering, and mathematics are NP-hard [19]. Some examples relevant for physics are finding the ground state energy of an Ising model [20], the training of variational quantum algorithms [21], or the quantum separability problem [22, 23], and many more.[1] These problems typically concern properties where all size parameters are bounded or even fixed. For example, the ground state energy problem concerns the minimal energy of Hamiltonians with fixed system size.

This highlights an analogy between certain classes of problems: an *unbounded problem* tests a property for an unbounded number of occurrences (which can be generated recursively), whereas the corresponding *bounded version* tests the same property for a bounded number of

---

[1]There are thousands of NP-complete problems. More than three hundred of them are presented in Ref. [24].

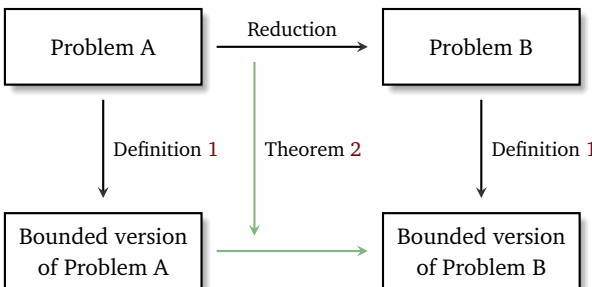

Fig. 1: If Problem B is at least as hard as Problem A (i.e. there is a reduction from A to B), is the bounded version of Problem B at least as hard as the bounded version of Problem A? Theorem 2 gives a sufficient condition when this is the case by reusing the reduction between their unbounded versions.

situations. This includes, for example, testing a certain property of a translational invariant spin system for all system sizes, or up to a certain size. One common observation in this context is that bounded versions of undecidable problems are NP-hard. This observation was already made in [9, 10, 25] for various examples, as well as in [19, Chapter 3].

Despite this analogy, the techniques used to prove NP-hardness and undecidability often differ. While proofs of undecidability mainly rely on reductions from the halting problem, the Post correspondence problem or the Wang tiling problem, NP-hardness proofs mainly rely on reductions from the satisfiability problem SAT, or from NP-complete graph problems like the 3-coloring problem or MAXCUT.

In this work, we establish a relation between undecidable problems and certain NP-hard problems. Specifically, we define a bounded version of a problem and a method to leverage the reduction from unbounded problems to their corresponding bounded problems (see Figure 1). Then we present two versions of the halting problem whose bounded versions are NP-hard, and use these, together with our method, to provide simple and unified proofs of the NP-hardness of the bounded version of the Post correspondence problem, the matrix mortality problem, the positivity of matrix product operators, the reachability problem, the tiling problem, and the ground state energy problem.

This work sheds light on the various intractability levels of problems used in theoretical physics by highlighting the computational consequences of bounding a parameter. More generally, this work is part of a tradition of studying problems from a computational perspective, which has proven extremely successful in mathematics and beyond [19]. For example, the hardness results of the ground state energy problem rule out a tractable solution of the ground state for a given Hamiltonian, both for unbounded system sizes as well as a fixed system size.

This paper is structured as follows. In Section 2, we present a definition of bounding and a method to leverage the reduction from unbounded problems to their corresponding bounded versions. In Section 3, we present the two halting problems which are the root undecidable problems. In Section 4, we apply our framework to many examples. In Section 5, we conclude and discuss future directions. The appendix contains basic background on computational complexity (Appendix A), the hardness proofs of the root undecidable problems (Appendix B) and more details on the discussed examples (Appendix C).

## 2 Bounding

In this section, we present a definition of a bounded version of a language (Section 2.1), and a method to leverage the reduction from unbounded problems to their corresponding bounded

versions (Section 2.2). For a short introduction to computational complexity, we refer the reader to Appendix A. To the best of our knowledge, no prior work introduces or studies bounded versions of problems from a general systematic perspective.

## 2.1 Definition of bounding

Let $\Sigma$ be a finite alphabet and $\Sigma^*$ the set of all words generated from $\Sigma$. A language $L \subseteq \Sigma^*$ encodes all the yes-instances of a given problem, i.e. $x \in L$ if $x$ is a yes-instance and $x \notin L$ if $x$ is a no-instance.

We now define a bounded version $L_B$ of $L$. For this purpose, we add a second parameter $n \in \mathbb{N}$ to every yes-instance in $L$. This parameter acts as an acceptance threshold for every yes-instance $x \in L$ and is encoded in unary, i.e. for $1 \in \Sigma$, every element of $L_B$ is of the form $\langle x, 1^n \rangle$, where $1^n$ represents the $n$-fold concatenation of 1.

**Definition 1.** *Let $L \subseteq \Sigma^*$ be a language. A language*

$$L_B \subseteq \left\{ \langle x, 1^n \rangle \mid x \in \Sigma^*, n \in \mathbb{N} \right\}$$

*is called a* bounded version *of $L$ if*

*(i)* $x \in L \iff \exists n \in \mathbb{N} : \langle x, 1^n \rangle \in L_B$.

*(ii)* $\langle x, 1^n \rangle \in L_B \implies \langle x, 1^{n+1} \rangle \in L_B$.

We shall often refer to $L$ as the *unbounded* language of $L_B$.

First, note that the definition of bounded versions relies only on the existence of a parameter $n$ in the problem that acts accordingly. While most problems we consider in this paper are RE-complete, Definition 1 applies to languages of arbitrary complexity. Moreover, note that the bounding parameter can also be encoded differently. For example, if the parameter is encoded in binary, most of the bounded version would be NEXP-hard instead of NP-hard. Finally, we remark that the process of bounding a language can be reversed. Given a language $L_B$ with instances of the form $\langle x, 1^n \rangle$ satisfying only Condition (ii), there is a unique language $L$, defined via (i), which is the unbounded language of $L_B$.

Many problems mentioned in the introduction contain a parameter that gives rise to a bounded version according to Definition 1. This parameter can be the system size for tensor network and spectral gap problems, or the dimension of the entangled state for quantum correlation scenarios; we will present many such examples in Section 4.

As an example, let us consider the halting problem HALT with its known bounded version BHALT. The former takes instances $\langle T, x_0 \rangle$ with a description $T$ of a Turing machine and an input $x_0$. An instance $\langle T, x_0 \rangle$ is accepted if and only if the Turing machine $T$ halts on $x_0$. The bounded halting problem takes instances $\langle T, x_0, 1^n \rangle$, which are accepted if and only if the Turing machine halts on $x_0$ within $n$ computational steps. BHALT is indeed a bounded version according to Definition 1 since halting of a Turing machine is equivalent to the existence of a finite halting time, and halting within $n$ steps implies halting within $n + 1$ steps.

We remark that in Definition 1 there is some freedom in the choice of the bounding parameter. For example, for every non-decreasing, unbounded function $f : \mathbb{N} \to \mathbb{N}$, the language

$$\text{BHALT}_f := \left\{ \langle T, x_0, 1^n \rangle \mid T \text{ halts on } x_0 \text{ in } f(n) \text{ steps} \right\}$$

is also a bounded version of HALT. In this paper, we will focus on the simplest versions setting $f = \text{id}$ in all examples.

## 2.2 Leveraging reductions to the bounded case

Given the hardness of the unbounded languages, what can we say about the bounded ones? We will now give a condition to leverage a reduction of unbounded problems to a reduction between the corresponding bounded problems. This results in a method to prove hardness results of many bounded versions of undecidable problems, as we will see in Section 4.

Let $L_B$ be a bounded version of $L \subseteq \Sigma^*$. For $x \in \Sigma^*$, we define the threshold parameter

$$n_{\min,L}[x] := \inf\{n \in \mathbb{N} : \langle x, 1^n \rangle \in L_B\},$$

where we set $\inf \emptyset = \infty$. In other words, $n_{\min}[x]$ denotes the minimum value of $n$ leading to an accepting instance of $L_B$. Note that $n_{\min}[x] < \infty$ for every $x \in L$ due to (i) of Definition 1 and $n_{\min}[x] = \infty$ if $x \notin L$. Moreover, $\langle x, 1^n \rangle \in L_B$ if and only if $n \geq n_{\min}[x]$ due to (ii) of Definition 1.

**Theorem 2.** *Let $L_1, L_2 \subseteq \Sigma^*$ be two languages and $\mathcal{R} : L_1 \to L_2$ a polynomial-time reduction from $L_1$ to $L_2$, i.e. $L_1 \leq_{\mathsf{poly}} L_2$. Furthermore, let $L_{B1}$ and $L_{B2}$ be bounded versions of $L_1$ and $L_2$, respectively.*

*If there is a strictly increasing polynomial $p : \mathbb{N} \to \mathbb{N}$ such that*

$$n_{\min,L_2}[\mathcal{R}(x)] \leq p\big(n_{\min,L_1}[x]\big), \tag{1}$$

*for every $x \in L$, then*

$$\langle x, 1^n \rangle \mapsto \langle \mathcal{R}(x), 1^{p(n)} \rangle \tag{2}$$

*is a polynomial-time reduction from $L_{B1}$ to $L_{B2}$, hence $L_{B1} \leq_{\mathsf{poly}} L_{B2}$.*

*Proof.* Since $\mathcal{R}$ and $p$ are polynomial-time maps, the map in Equation (2) is also polynomial-time. It remains to show that yes/no-instances are preserved via this map. We have that $\langle x, 1^n \rangle \in L_{B1}$ if and only if $n \geq n_{\min,L_1}[x]$. This is equivalent to

$$p(n) \geq p\big(n_{\min,L_1}[x]\big) \geq n_{\min,L_2}[\mathcal{R}(x)],$$

since $p$ is a strictly increasing function. But this is again equivalent to $\langle \mathcal{R}(x), 1^{p(n)} \rangle \in L_{B2}$. $\qquad\square$

In words, Condition (1) demands that there is a polynomial that relates thresholds of $x$ and $\mathcal{R}(x)$ for all $x$. We require that $p$ is strictly increasing instead of mere non-decreasing as we need the equivalence of the statements $n \geq m$ and $p(n) \geq p(m)$ in the proof.

Many known reductions of undecidable problems implicitly contain such a polynomial $p$ in their construction. This gives an almost-for-free proof of the NP-hardness of their bounded problems. However, most of these works do not make this polynomial explicit and therefore do not obtain the NP-hardness results. While the theorem only assumes that $p(n_{\min,L_2}[x])$ upper bounds $n_{\min,L_1}[\mathcal{R}(x)]$, in all examples, we have an equality between these expressions. In Section 4, we will present many examples of this behavior.

Theorem 2 also generalizes to other types of reductions. For example, we obtain an exponential-time reduction between the bounded versions when $\mathcal{R}$ is considered a exponential-time reduction and $p$ being a strictly increasing function that can be computed in exponential time.

# 3 Halting problems as root problems

The result of Theorem 2 gives only relative statements about hardness. Specifically, it allows to construct a reduction between bounded versions given a reduction between their original

problems. To prove NP/coNP-hardness of bounded problems, we need root problems with bounded versions whose complexities are already known. In this section, we review two fundamental undecidable problems and their bounded versions, namely two variants of the halting problem.

While HALT and BHALT are the most basic versions of halting problems, we need variations of the halting problem that take non-deterministic Turing machines as inputs. This is due to the fact that, while HALT is undecidable, BHALT is in P. Since we want to prove NP/coNP-hardness of bounded problems, we need root problems with a NP/coNP-hard bounded version to start the reduction from. Therefore, we introduce two non-deterministic versions of HALT, called NHALT and NHALTALL, with an NP-hard and a coNP-hard bounded version, respectively.

(a) The problem NHALT checks the halting of a non-deterministic Turing machine on the empty tape. An instance is given by a description of a non-deterministic Turing machine $T$, which is accepted if and only if $T$ halts on the empty tape.[2] Its bounded version BNHALT takes instances $\langle T, 1^n \rangle$ and accepts if and only if $T$ halts on the empty tape in at most $n$ steps. The unbounded problem is RE-hard since it contains the (deterministic) halting problem on the empty tape, which is itself RE-hard. Its bounded version BNHALT is NP-hard.

(b) The problem NHALTALL takes a description of a non-deterministic Turing machines $T$ as an instance, which is accepted if and only if $T$ halts on the empty tape along *all* computation paths. Its bounded version BNHALTALL is given by instances $\langle T, 1^n \rangle$ which are accepted if and only if $T$ halts on the empty tape within $n$ computational steps along *all* computation paths. The unbounded problem is RE-hard, and the bounded version is coNP-hard.

For more details on these problems and their complexity proofs, we refer to Appendix B.

NHALT will be the root problem to prove the hardness of the bounded Post correspondence problem (Section 4.1) and the bounded matrix mortality problem (Section 4.2). NHALTALL will be the root problem to prove the hardness of the bounded Tiling problem (Section 4.7).

While reductions for undecidable problems usually stem from the deterministic halting problem HALT, here we need non-deterministic halting problems in order to prove NP-hardness of the bounded versions. Canonical extensions of the reductions from HALT to a non-deterministic halting problem lead to different choices of root problems. For example, the Post correspondence problem has a similar structure as NHALT, while the structure of the tiling problem relates to NHALTALL. We will elaborate on these structures in the corresponding sections.

We expect that other variants of the halting problem serve as root problems for other complexity results; see Section 5 for further discussion.

# 4 A tree of undecidable problems and their bounded versions

In this section, we apply Theorem 2 to several undecidable problems in order to prove the NP-hardness of the bounded versions. The problems studied in this paper are summarized in Figure 2, where every edge corresponds to one application of the theorem. For a detailed treatment of these problems and further examples, we refer to Appendix C.

## 4.1 The Post Correspondence Problem

The Post correspondence problem (PCP) [26] is an undecidable problem with a particularly simple and intuitive formulation. For this reason, it is often used to prove undecidable results

---

[2]In other words, it accepts if and only if there is a computation path such that $T$ halts along this path.

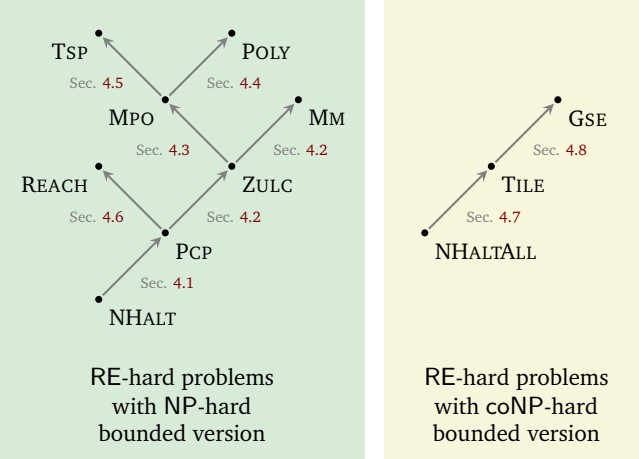

Fig. 2: The problems and reductions considered in this work. NHALT is the non-deterministic halting problem, PCP is the Post correspondence problem, REACH is the reachability problem for resource theories, ZULC is the zero in the upper left corner problem, MM is the matrix mortality problem, MPO is the positivity of Matrix product operators problem, TSP is the stability of positive maps problem and POLY is the polynomial positivity problem. NHALTALL is the non-deterministic halting problem on all computational paths, TILE is the Wang tiling problem, and GSE is the ground state energy problem. NHALT and NHALTALL are the root problems, and every arrow corresponds to a reduction, explained in the referenced subsection.

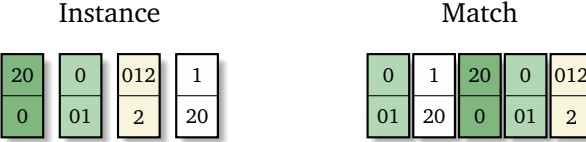

Fig. 3: An instance of PCP is a set of dominos (left). This is a yes-instance if they form a match (right), i.e. the words on the top and the bottom coincide.

in quantum information theory [12], including a version of the matrix product operator positivity problem [9], threshold-problems for probabilistic and quantum finite automata [15], or reachability problems in resource theories [16]. It is stated as follows:

*Given two finite sets of words, $\{a_1, \ldots, a_k\}$ and $\{b_1, \ldots, b_k\} \subseteq \Sigma^*$, is there a finite sequence of indices $i_1, \ldots, i_\ell$ such that*

$$a_{i_1} a_{i_2} \ldots a_{i_\ell} = b_{i_1} b_{i_2} \ldots b_{i_\ell} \ ?$$

This decision problem is known to be RE-complete via a reduction from the halting problem. Since $a_i$ and $b_i$ only appear in fixed pairs, this problem has an equivalent formulation in terms of dominoes

$$d_i = \left[ \frac{a_i}{b_i} \right].$$

The question is whether there exists a finite arrangement of dominoes that form a match, i.e. where the upper and lower parts coincide when the words are read across the dominoes (see Figure 3).

We define a bounded version of PCP that checks for sequences of length at most $n$:

*Given a finite set of dominoes $\{d_1, \ldots, d_k\}$ and a number $n \in \mathbb{N}$ in unary, is there a matching arrangement of dominoes $d_{i_1}, \ldots, d_{i_\ell}$ with $\ell \leq n$?*

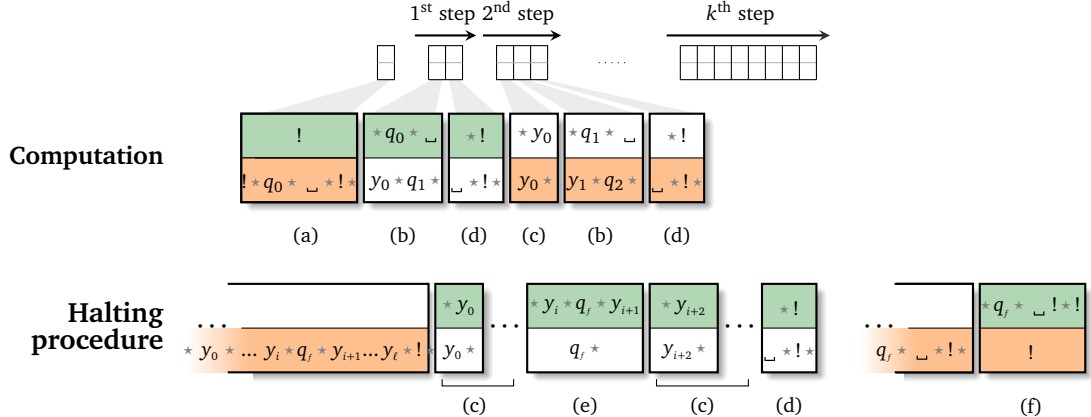

Fig. 4: (Top) In the reduction NHALT → PCP, domino (a) contains the initial configuration of the TM, i.e. an empty tape with head at position zero. Each computation step is simulated by copying the lower string to the upper part in green. This is done by applying a transition domino (b), reproducing the tape (c), and adding a new empty tape slot (d). This generates a new string on the bottom, showing the new instantaneous description (white). Repeating the procedure simulates the computation. (Bottom) The halting of the Turing machine is mapped to the following match of tiles. When the Turing machine reaches the final state $q_f$, the instantaneous description is successively removed by dominoes (e). Adding a final domino (f) guarantees the match.

This problem, denoted BPCP, is a bounded version of PCP according to Definition 1. It is known to be NP-complete (see [9,24,27] for the ideas of the reductions). The basic idea of the reduction is analogous to Theorem 2, i.e. using the reduction of the (unbounded) undecidable problems to relate the bounding parameters via a polynomial-time map. Yet, the usual reductions do not directly give rise to a polynomial relation between the bounding parameters, contrary to what is claimed in [9]. We will now provide a reduction NHALT → PCP leading to such a relation (and refer to Appendix C.1 for further details). Our approach is similar to that of [28].

We define a map $\mathcal{R}$ mapping a description of a Turing machine to a set of dominoes, $\mathcal{R}(T) := \langle d_1, \ldots, d_k \rangle$. This map mimics the description of $T$ (see Figure 4). For example, $d_1$ is a domino whose lower string is given by

$$! \star q_0 \star \sqcup \star ! \star,$$

where ! and $\star$ are separator symbols, and $q_0$ and $\sqcup$ indicate that the Turing machine head is initially in state $q_0$ acting on an empty tape.

The map $\mathcal{R}$ is a polynomial-time map; in particular, the number of dominoes $k$ is polynomial in the description size of $T$. From the construction of $\mathcal{R}$ it follows that $T$ halts on the empty tape if and only if there exists a match of dominoes $d_1, \ldots, d_k$. This implies that $\mathcal{R}$ is a polynomial-time reduction from the non-deterministic halting problem, which implies that PCP is RE-hard.

Refining this argument and using Theorem 2, we obtain that $\mathcal{R}$ can be used as a reduction from BNHALT to BPCP. Each computation step of $T$ on the empty tape is simulated by attaching dominoes, as shown in Figure 4. This procedure guarantees that $T$ halts within $n$ steps if and only if $d_1, \ldots, d_k$ form a match within

$$p(n) := (n+1) \cdot (n+2)$$

steps. Hence, the halting time of $T$ is polynomially related to the length of a minimal match of $\mathcal{R}(T)$. This proves that BPCP is NP-hard by Theorem 2.

Moreover, PCP is RE-complete and BPCP is NP-complete, by taking matching domino arrangements as certificates, and a polynomial-time verifier that checks arrangements.

## 4.2 The Zero in the upper left corner and Matrix Mortality problem

We now present the Matrix mortality problem (MM) and the zero in the upper left corner problem (ZULC) with their bounded versions. Both problems are undecidable and have been applied to prove the undecidability of quantum information problems such as the positivity of Matrix product operators [8] (see Section 4.3), the reachability problem [12] (see Section 4.6), or the measurement occurrence problem [11].

The matrix mortality problem is the following:

*Given $A_1, \ldots, A_k \in \mathcal{M}_d(\mathbb{Q})$, is there a finite sequence $i_1, \ldots, i_\ell \in \{1, \ldots, k\}$ such that*

$$A_{i_1} \cdot A_{i_2} \cdots A_{i_\ell} = \mathbf{0} \ ?$$

Here, $\mathbf{0}$ denotes the zero matrix, and $\mathcal{M}_d(\mathbb{Q})$ the set of $d \times d$ matrices over $\mathbb{Q}$. ZULC is almost identical to MM, the only difference is that only the upper left corner of the product $A_{i_1} \cdot A_{i_2} \cdots A_{i_\ell}$ is asked to be zero. We define the bounded matrix mortality problem (BMM) and the bounded zero in the upper left corner problem (BZULC) by adding a parameter $n \in \mathbb{N}$ to every instance, and asking whether the desired zeros can be realized within $n$ matrix multiplications.

The undecidability of MM was first proven by Paterson [29]. Since then, many tighter bounds on the number and size of matrices for both problems have been found (see [30] and references therein). It is also known that BMM is NP-hard [25]. However, the proof relies on a reduction from the NP-complete problem SAT and is therefore independent of the original reduction proving undecidability. To the best of our knowledge, the following is the first proof of the NP-hardness of these bounded matrix problems using the same reductions as their unbounded versions.

We briefly sketch the reductions and refer to Appendix C.3 for details. Following [31], there exist polynomial-time reductions $\mathcal{R} : \text{PCP} \to \text{ZULC}$ and $\mathcal{Q} : \text{ZULC} \to \text{MM}$ with the following properties:

(i) The dominoes $\mathbf{d} := \langle d_1, \ldots, d_k \rangle$ form a match of length $n$ if and only if the matrices

$$\langle N_1, \ldots, N_{k'} \rangle := \mathcal{R}(\mathbf{d})$$

multiply to a matrix with a zero in the upper left corner within $n$ matrix multiplications.

(ii) The matrices $\mathbf{N} := \langle N_1, \ldots, N_\ell \rangle$ form a zero in the upper left corner using $n$ matrix multiplications if and only if the matrices

$$\langle M_1, \ldots, M_{\ell'} \rangle := \mathcal{Q}(\mathbf{N})$$

multiply to a zero matrix within $n + 2$ matrix multiplications.

Together with Theorem 2, these observations show that

$$\langle x, 1^n \rangle \mapsto \langle \mathcal{R}(x), 1^n \rangle$$

is a polynomial-time reduction from BPCP to BZULC, and

$$\langle x, 1^n \rangle \mapsto \langle \mathcal{Q}(x), 1^{n+2} \rangle$$

is a polynomial-time reduction from BZULC to BMM. This proves that BZULC and BMM are NP-hard.

Moreover, MM and ZULC are RE-complete, and their bounded versions, BMM and BZULC, are NP-complete by taking matching matrix arrangements as certificates and a polynomial-time verifier checking the statements.

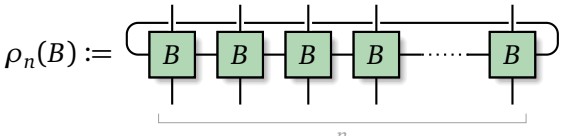

Fig. 5: Tensor network representation of the MPO $\rho_n(B)$. The MPO problem asks: Given a tensor $B$, is $\rho_n(B)$ positive semidefinite for all $n$?

## 4.3 The matrix product operator (MPO) positivity problem

A matrix product operator (MPO) representation is a decomposition of a multipartite operator into local tensors according to a one-dimensional structure [32,33]. A local tensor $B$ defines a diagonal operator $\rho_n(B)$ for every system size $n$ (see Figure 5). More precisely, given a family of $D \times D$ matrices $(B_i)$ for $i \in \{1, \dots, d\}$, the diagonal (translationally invariant) MPO of size $n$ is given by

$$\rho_n(B) := \sum_{i_1, \dots, i_n = 1}^{d} \mathrm{tr}\left(B_{i_1} \cdots B_{i_n}\right) |i_1 \dots i_n\rangle \langle i_1 \dots i_n| \,.$$

If these MPO should represent density matrices, then $B$ should be such that $\rho_n(B)$ is positive semidefinite for every $n$. This property cannot be decided algorithmically, not even for classical states. In other words, the following MPO problem is undecidable:

*Given $B_1, \dots, B_k \in \mathcal{M}_D(\mathbb{Q})$, is there $n \in \mathbb{N}$ such that $\rho_n(B) \ngeq 0$?*

Note that an MPO is usually defined more generally; instead of restricting to families of diagonal (classical) matrices $B_i$, a general matrix product operator is defined via families of $D \times D$ matrices $(B_i^j)$ for $i, j = 1, \dots, d$, addressing also non-diagonal entries of the matrix. However, as diagonal MPOs are contained in this definition, the undecidability of MPO as we defined it implies that the same problem for arbitrary matrix product operators is also undecidable.

Similar to previous bounded versions, we define BMPO by bounding the system size $n$:

*Given $B_1, \dots, B_k$ and $n \in \mathbb{N}$, is there an $\ell \le n$ such that $\rho_\ell(B) \ngeq 0$?*

Note that MPO is usually stated in the negated way; yet, we use this definition for consistency with the definition of bounding.

Let us now sketch the proof that BMPO is NP-hard by using Theorem 2 together with the undecidability proof of [8]. For details, we refer to Appendix C.4. Following [8], every instance $\langle A_1, \dots, A_k \rangle$ of ZULC is mapped to $k+1$ matrices $B_1, \dots, B_{k+1} \in \mathcal{M}_D(\mathbb{Z})$. These matrices are constructed such that $(B_{i_1} \cdots B_{i_\ell})_{11} = 0$ if and only if $\exists j_1, \dots, j_{\ell+1} \in [k+1]$ such that

$$\mathrm{tr}\left(B_{j_1} \cdots B_{j_\ell} \cdot B_{j_{\ell+1}}\right) < 0 \,.$$

This implies that the family $A_1, \dots, A_k$ generates a zero in the upper left corner using $n$ matrix multiplications if and only if $\rho_{n+1}(B) \ngeq 0$. Setting $p(n) = n + 1$ proves that BMPO is NP-hard by Theorem 2.

Moreover, MPO is RE-complete and BMPO is NP-complete by defining negative diagonal entries as certificates.

While MPO precisely characterizes psd matrix product operators, in practice, algorithms distinguishing MPOs that are sufficiently positive or that violate positivity by at least an error $\varepsilon > 0$ are often acceptable. This is the idea behind weak membership problems. Along these lines, we define the approximate MPO problem $\mathrm{MPO}_\varepsilon$ as follows:

Given $C_1, \dots, C_k \in \mathcal{M}_D(\mathbb{Q})$ with $\mathrm{tr}(\rho_\ell(C)) \leq 1$ for every $\ell \in \mathbb{N}$ and a family of errors $(\varepsilon_\ell)_{\ell \in \mathbb{N}}$ with $0 < \varepsilon_\ell \leq 1/\exp(\ell)$. Decide the following:

(a) Accept if $\exists n \in \mathbb{N} : \rho_n(C) \not\geq -\varepsilon_n \mathbb{1}$.

(b) Reject if $\forall n \in \mathbb{N} : \rho_n(C) \geq \varepsilon_n \mathbb{1}$.

$\mathrm{MPO}_\varepsilon$ is undecidable using the same reduction as above and the fact that $\mathrm{tr}(\rho_n(C))$ increases exponentially in $n$ in the above reduction. Following the usual bounding process, we define $\mathrm{BMPO}_\varepsilon$ by bounding $n$:

Given $C_1, \dots, C_k \in \mathcal{M}_D(\mathbb{Q})$ with $\mathrm{tr}(\rho_\ell(C)) \leq 1$ for every $\ell \in \mathbb{N}$, a family of errors $(\varepsilon_\ell)_{\ell \in \mathbb{N}}$ with $0 < \varepsilon_\ell \leq 1/\exp(\ell)$ and $n \in \mathbb{N}$. Decide the following:

(a) Accept if $\exists \ell \leq n : \rho_\ell(C) \not\geq -\varepsilon_n \mathbb{1}$.

(b) Reject if $\forall \ell \leq n : \rho_\ell(C) \geq \varepsilon_n \mathbb{1}$.

It follows that $\mathrm{BMPO}_\varepsilon$ is a bounded version of $\mathrm{MPO}_\varepsilon$ according to Definition 1. Moreover, Theorem 2 implies that $\mathrm{BMPO}_\varepsilon$ is also NP-hard.

We remark that Kliesch et al. [9] present a similar idea, by constructing a reduction from PCP to an alternative version of MPO and bounding both problems.

## 4.4 The polynomial positivity problem

The undecidability of MPO leads to the undecidability of other positivity problems. One of them concerns deciding the positivity of a certain class of polynomials [13]:

Given a family of polynomials $q_{\alpha,\beta}(\underline{x})$ for $\alpha, \beta \in \{1, \dots, D\}$ with integer coefficients, is there an $n \in \mathbb{N}$ such that the polynomial

$$p_n(\underline{x}_1, \dots, \underline{x}_n) := \sum_{\alpha_1, \dots, \alpha_n = 1}^{D} q_{\alpha_1, \alpha_2}(\underline{x}_1) \cdots q_{\alpha_n, \alpha_1}(\underline{x}_n) \tag{3}$$

is not nonnegative (i.e. $p_n(\mathbf{a}) < 0$ for some $\mathbf{a} \in \mathbb{R}^{d \cdot n}$)?

Here $\underline{x}_i$ denotes a $d$-tuple of variables, for every $i$. We define this problem as POLY and its bounded version (by restricting to checking nonnegativity of $p_k$ for $k \leq n$) by BPOLY.

Following the proof of [13], POLY is RE-hard since there exists a polynomial-time map

$$\mathcal{R}(\langle B_1, \dots, B_k \rangle) := \langle q_{\alpha,\beta} : \alpha, \beta = 1, \dots, D \rangle,$$

such that

$$\rho_n(B) \geq 0 \text{ if and only if } p_n \text{ is nonnegative.}$$

This implies that $\langle B, 1^n \rangle \mapsto \langle \mathcal{R}(B), 1^n \rangle$ defines a reduction from BMPO to BPOLY. It follows that BPOLY is NP-hard. We refer to Appendix C.5 for further details.

## 4.5 Stability of positive maps

Another undecidable problem related to positivity concerns tensor products of positive maps. A map

$$\mathcal{P} : \mathcal{M}_d(\mathbb{C}) \to \mathcal{M}_d(\mathbb{C})$$

is called *positive* if it maps positive semidefinite matrices to positive semidefinite matrices. Such a map is called *n-tensor-stable positive* if $\mathcal{P}^{\otimes n}$ is a positive map, and *tensor-stable positive* if it is

$n$-tensor-stable positive for all $n \in \mathbb{N}$. The existence of non-trivial tensor-stable positive maps relates to the existence of NPT bound-entangled states [34].

Let us define the $n$-fold Matrix Multiplication tensor as

$$|\chi_n\rangle := \sum_{i_1,\ldots,i_n=1}^{s} |i_1,i_2\rangle \otimes |i_2,i_3\rangle \otimes \cdots \otimes |i_n,i_1\rangle \,,$$

and denote the projection to this vector by

$$\chi_n := |\chi_n\rangle\langle\chi_n| \,. \tag{4}$$

The following problem is undecidable [14]:

> *Given a positive map $\mathcal{P} : \mathcal{M}_d(\mathbb{C}) \to \mathcal{M}_d(\mathbb{C})$, is $\mathcal{P}^{\otimes n}(\chi_n)$ not positive semidefinite for some $n \in \mathbb{N}$?*

We denote this problem by TSP. Its bounded version, BTSP takes instances $\langle \mathcal{P}, 1^n \rangle$ and asks the same question for $k$-fold tensor products with $k \le n$.

From the proof of [14] it follows that there is a polynomial-time map $\langle B_1, \ldots, B_k \rangle \mapsto \mathcal{P}$ such that the resulting $\mathcal{P}$ satisfies that $\mathcal{P}^{\otimes n}(\chi_n) = \rho_n(B)$. This shows that TSP is RE-hard by a reduction from MPO. Moreover, since

$$\rho_n(B) \geqslant 0 \quad \text{if and only if} \quad \mathcal{P}^{\otimes n}(\chi_n) \geqslant 0 \,,$$

it follows BTSP is NP-hard by applying Theorem 2 together with the fact that BMPO is NP-hard. We refer to Appendix C.6 for more details on the reduction.

## 4.6 The reachability problem in quantum information

The reachability problem in quantum information concerns the question whether a resource state $\rho$ (given as a density matrix) can be converted to another state $\sigma$ by using only free resource operations from a fixed set $\mathcal{F} := \{\Phi_1, \ldots, \Phi_k\}$. More precisely, we define REACH as follows:

> *Given density matrices $\rho$, $\sigma \in \mathcal{M}_d(\mathbb{C})$ and a set $\mathcal{F}$ of free operations $\mathcal{M}_d(\mathbb{C}) \to \mathcal{M}_d(\mathbb{C})$, is there a map*
> $$\Phi := \Phi_{i_n} \circ \Phi_{i_{n-1}} \circ \cdots \circ \Phi_{i_1}$$
> *in the free semigroup $\mathcal{F}^*$ such that $\sigma = \Phi(\rho)$?*

The free semigroup $\mathcal{F}^*$ of $\mathcal{F}$ consists of all maps generated by finite compositions of maps in $\mathcal{F}$. We denote by $\mathcal{F}^n$ the set of all operations arising from at most $n$ compositions of maps in $\mathcal{F}$, and define the bounded version BREACH by replacing $\mathcal{F}^*$ with $\mathcal{F}^n$ in the above problem statement.

REACH is undecidable via a reduction from PCP [16]. We now prove that the bounded version BREACH is NP-hard. We rely on Scandi and Surace's work [16], who provide a polynomial-time reduction $\mathcal{R}$ mapping dominoes $d_i$ to two types of resource maps $H_i^\lambda, G_i^\lambda$ for $\lambda \in (0,1)$. The set of free resource operations is then specified by

$$\mathcal{F} = \left\{ \mathbb{1}, H_i^\lambda, G_i^\lambda : i = 1, \ldots, r \text{ and } \lambda \in (0,1) \right\}.$$

For a state $\rho \in \mathcal{M}_4(\mathbb{C})$, it is shown that

$$\sigma := \lambda\rho + (1-\lambda)\frac{\mathbb{1}}{4}$$

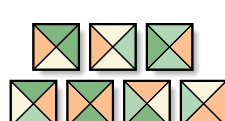
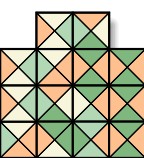

Instance        Valid tiling

Fig. 6: An instance of TILE is a set of tiles (left). A set of tiles is a yes-instance if there exists a valid tiling of the plane. Part of a potentially valid tiling is shown on the right. In a valid tiling, the colors of adjacent tiles must coincide and the tiles cannot be rotated.

is reachable via operations in $\mathcal{F}^*$ if and only if there exists a match of the corresponding dominoes in PCP. This shows that REACH is RE-hard. More specifically, there exists a match of length $n$ if and only if

$$\sigma = G_{i_n}^{\lambda_n} \circ \cdots \circ G_{i_1}^{\lambda_1} \circ H_{i_1}^{\lambda_1} \circ \cdots \circ H_{i_n}^{\lambda_n}(\rho),$$

for a choice $\lambda_1, \ldots, \lambda_n \in (0, 1)$. In other words, a threshold parameter $n$ in BPCP is mapped to a threshold $2n$ in BREACH. This proves that BREACH is NP-hard by applying Theorem 2.

## 4.7 The tiling problem

Let us now consider the Wang tiling problem. This problem has been used to prove undecidability in many physics-related problems, like the spectral gap problem in 2D [1], 2D PEPS problems [10], or the universality of translational invariant, classical spin Hamiltonians in 2D [35].

A tile is given by a square with different colors on each side of the tile (see Figure 7). Given a finite set of tiles, a valid tiling is an arrangement of tiles whose adjacent edges coincide. Moreover, all tiles have a fixed orientation, i.e. they cannot rotate. We study the following variant:

*Given a set of tiles $\mathcal{T} = \{t_1, \ldots, t_k\}$, is it impossible to tile the plane when $t_1$ is in the origin?*

Note that this problem is usually stated in the negated form, but this formulation is more convenient for our purposes. The constraint on the fixed tile in the origin can also be removed [36, 37]; we stick to this version for simplicity. The corresponding bounded version is the following:

*Given a set of tiles $\mathcal{T} = \{t_1, \ldots, t_k\}$ and $n \in \mathbb{N}$, is it impossible to tile $\mathbb{Z}_n^2$ when $t_1$ is in the origin?*

Here we denote by $\mathbb{Z}_n^2 := \{-n, \ldots, 0, \ldots, n\}^2$ the square grid of size $(2n + 1) \times (2n + 1)$ around the origin.

Let us now sketch the proof that TILE is RE-hard and that BTILE is coNP-hard. This will imply that the tiling problem in its usual formulation ("can the plane be tiled?") is coRE-hard and its bounded version is NP-hard. We refer to Appendix C.7 for details.

In contrast to the previous examples, we now construct a reduction from NHALTALL instead of NHALT. While to check whether $\{d_1, \ldots, d_k\}$ is a yes-instance of BPCP, one needs to find a *single* matching arrangement, to verify whether $\{t_1, \ldots, t_k\}$ is a yes-instance of BTILE one has to check (for a fixed size $n$) whether *all* arrangements of tiles in $\mathbb{Z}_n^2$ are invalid. This structure is similar to NHALTALL, where for a fixed computation time $n$, one needs to check whether a given Turing machine $T$ halts on *all* computation steps. More precisely, there is a polynomial relation between the bounding parameters of BTILE and BNHALTALL, as needed for Theorem 2.

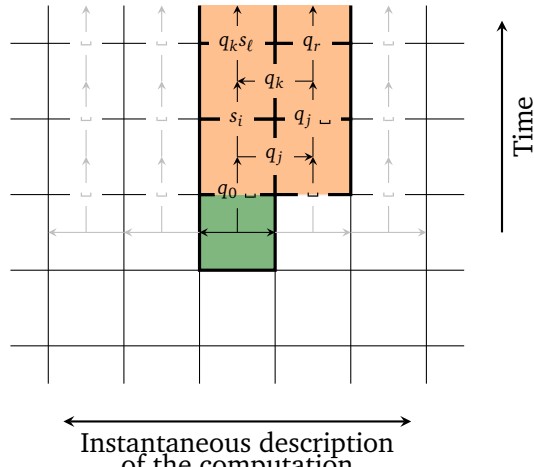

Instantaneous description
of the computation

Fig. 7: In the reduction NHALTALL → TILE, the instantaneous description of the Turing machine is mapped to a horizontal configuration of tiles, and every computational step is mapped to a valid tiling of the horizontal line above. The green tile is fixed at the origin, while the orange tiles realize the computation. The rest of the plane is filled with trivial tiles, such as the empty tiles (bottom) or tiles copying the tape information (left and right). A Turing machine halts along every path within $n$ steps if and only if the corresponding tiling terminates after $n$ horizontal lines.

We build a polynomial-time reduction from NHALTALL to TILE following [37]. The reduction maps a description of a Turing machine $T$ to a set of tiles representing either a slot in the tape or a computational step. The (infinite) starting tape is mapped to the fixed origin tile representing the empty tape with head position at zero. Filling up a new line corresponds to one computational step. This reduction also applies to non-deterministic Turing machines.

The reduction is such that the tiling cannot be continued after filling up $n$ lines if and only if $T$ halts on all computation paths after at most $n$ computational steps. We refer to Figure 7 and Appendix C.7 for further details on the reduction. This proves that TILE is undecidable. By Theorem 2, we obtain that BTILE is coNP-hard, since the maximal halting time $n$ on every computation path is mapped to the termination size $n + 1$.

In addition, TILE is RE-complete by taking a system size where all tilings terminate as a certificate and an exponential-time verifier checking all tilings of this size. BTILE is coNP-complete by choosing tilings as a certificate and a polynomial-time verifier checking the validity of the tiling. This highlights that when proving completeness, *not* every construction in the unbounded case trivially translates to the bounded version.

## 4.8 Ground State Energy problem

We now study a version of the ground state energy problem. For this purpose, we consider a spin system on a 2D grid. We assume that every spin takes values in a set $\mathcal{S}$. Given coupling functions $h^x, h^y : \mathcal{S} \times \mathcal{S} \to \mathbb{N}$ and a local field $h^{\mathrm{loc}} : \mathcal{S} \to \mathbb{N}$, we define the Hamiltonian

$$H_n(\mathbf{s}) = h^{\mathrm{loc}}(s_{00}) + \sum_{\langle \mathbf{a}, \mathbf{b} \rangle_x} h^x(s_{\mathbf{a}}, s_{\mathbf{b}}) + \sum_{\langle \mathbf{a}, \mathbf{b} \rangle_y} h^y(s_{\mathbf{a}}, s_{\mathbf{b}}),$$

where $\mathbf{s} = (s_{ij})_{i,j \in \{-n,...,0,...,n\}}$ is a given spin configuration on the grid $\mathbb{Z}_n^2$ taking values in $\mathcal{S}$ and $s_{\mathbf{a}}, s_{\mathbf{b}}$ denote the elements with coordinates $\mathbf{a}$ and $\mathbf{b}$ in this array. Moreover, $\langle \mathbf{a}, \mathbf{b} \rangle_{x/y}$ denotes all neighbors in $x/y$-direction on $\mathbb{Z}_n^2$ where the $\mathbf{a}$ has a smaller $x/y$-coordinate than $\mathbf{b}$. Hence, $H_n$ is translational invariant except for the local field on the spin in the origin.

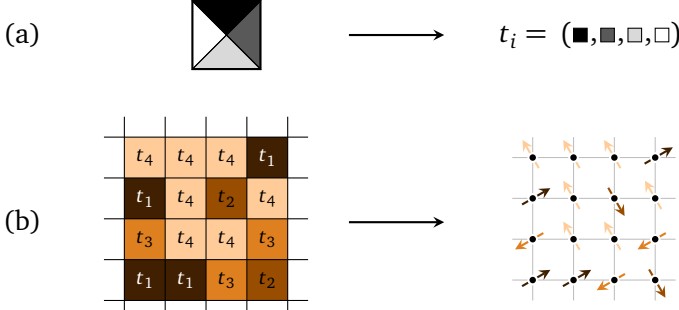

Fig. 8: In the reduction TILE → GSE, (a) every tile $t_i$ is mapped to a spin state $s_i$. (b) Every (valid and invalid) tiling maps to a spin configuration. A tiling of size $n$ is valid iff the corresponding spin configuration is the ground state of $H_n$ with energy 0.

We start by defining the bounded version of this problem, namely the bounded ground state energy problem BGSE:

*Given system size $n \in \mathbb{N}$, non-negative functions $h^x, h^y, h^{loc}$ and energy $E \in \mathbb{Q}$, is the ground state energy $E_{\min}(H_n) > E$?*

A function $h$ is non-negative if it is non-negative on its whole domain. Note that BGSE is indeed a bounded version, as $E_{\min}(H_{n+1}) \geq E_{\min}(H_n) > E$ since all couplings are non-negative. Further note that BGSE is usually formulated in the negated way, i.e. the question is if there exists a spin configuration whose energy is below the threshold $E$.

We now extend BGSE to an unbounded ground state energy problem GSE:

*Given non-negative functions $h^x, h^y, h^{loc}$ and an energy $E \in \mathbb{Q}$, is there an $n \in \mathbb{N}$ such that $E_{\min}(H_n) > E$?*

Note that BGSE is the bounded version of GSE according to Definition 1.

Let us show that GSE is RE-hard and BGSE is coNP-hard by a reduction $\mathcal{R} : \text{TILE} \to \text{GSE}$ (see Figure 8). Given a set of tiles $\mathcal{T} = \{t_1, \ldots, t_k\}$, we define the set of spin states as the set of tiles $\mathcal{S} := \mathcal{T}$. Since each tile is specified by four colors in a color space $C$, it can be represented as a 4-tuple

$$t_i = \left( t_i^N, t_i^E, t_i^S, t_i^W \right),$$

where the entries represent the colors on the top, right, bottom, and left of the tile. We define the coupling function so that a valid tiling with $t_1$ in the origin maps to a spin configuration of energy 0, and every inconsistent color pairing in an invalid tiling gives an additional energy penalty of 1. More precisely,

$$h^x(s, \hat{s}) := 1 - \delta(s^E, \hat{s}^W) \ \text{ and } \ h^y(s, \hat{s}) := 1 - \delta(s^N, \hat{s}^S),$$

where $s, \hat{s} \in \mathcal{S}$. According the definition of $H_n$, the first component of $h^x$ addresses the spin on the left and the second the spin on the right while the first component of $h^y$ addresses the spin on the bottom and the second the spin on the top. Moreover, we define

$$h^{\text{loc}}(s) := 1 - \delta(s, t_1).$$

Note that $H_n$ has a ground state of energy zero if and only if there exists a valid tiling of $\mathbb{Z}_n^2$ with tile $t_1$ at the origin. That is, $E_{\min}(H_n) > 0$ if and only if there is no valid tiling of size $n$. This guarantees that $\mathcal{R}$ is a reduction. Additionally, we obtain a reduction from BTILE to BGSE since the bounding parameters are identical. Similar to the tiling problem, one can show that GSE is RE-complete and BGSE is coNP-complete.

Note that non-translational invariant versions of BGSE are known to be coNP-hard since their negated versions are NP-hard. In particular, the ground state energy problem for 2D Ising models with fields is NP-complete [20].

## 5  Conclusions and Outlook

In this work, we have shown a relation between the hardness of an (unbounded) problem and the hardness of its bounded version. In particular, we have defined a bounded version of a language (Definition 1) and given a condition under which a reduction between the unbounded problems translates to a reduction between their bounded versions (Theorem 2). We have also applied this result to two classes of examples (Section 4): First, we showed that RE-hard problems like PCP, MPO, or REACH have an NP-hard bounded version; Second, we showed that RE-hard problems like TILE and GSE have a coNP-hard bounded version.

It would be interesting to extend this work to problems in quantum physics such as the spectral gap problem [1,2] or membership problems for quantum correlations [3–7]. A bounded version of the latter uses the dimension of the entangled state as the bounding parameter.

Another open question is whether the undecidability of Diophantine equations [38] and the NP-hardness of its bounded version [39] fits into our framework.[3] In this context, the unbounded problem is as follows:

*Given a Diophantine equation $p(\mathbf{x}, \mathbf{y}) = 0$ with 2k variables, and a k-tuple of integers $\mathbf{a} \in \mathbb{Z}^k$, does there exist $\mathbf{b} \in \mathbb{Z}^k$ such that $p(\mathbf{a}, \mathbf{b}) = 0$?*

Note that here $k$ is fixed. The bounded version would restrict to values $\mathbf{b} \in \{-n, \ldots, n\}^k$, where $n$ acts as the bounding parameter.

Are there also hard bounded versions with other types of complexity, such as QMA-hard [40] bounded versions? While we only considered the scenario of RE-hard problems with either NP-hard or coNP-hard bounded versions, there might be "root problems" whose bounded version is neither NP-hard or coNP-hard. Natural candidates for QMA-hard bounded version are the bounded/unbounded satisfiability problems of quantum circuits [41], which concerns Turing machines generating polynomial-size quantum circuits. The results of this work would imply that certain QMA-hard problems, like the ground state energy problem for $k$-local quantum Hamiltonians [42], relate to unbounded problems which are undecidable.

Finally, is it possible to prove the converse direction of Theorem 2? Since bounded languages give rise to a unique unbounded language, can every reduction between bounded versions be transferred to a reduction between the corresponding unbounded problems? If the bounded reduction is of the special form

$$\mathcal{R}_b : \langle x, n \rangle \mapsto \langle \mathcal{R}(x), p(n) \rangle,$$

with $p$ being a strictly increasing polynomial, then $\mathcal{R}$ is automatically a reduction between the unbounded problems. Yet, the question is open for general $\mathcal{R}_b$.

## 6  Acknowledgments

This project originated at a group retreat together with the group of Hans Briegel and Thomas Müller in Obergurgl, and we thank all participants for the fruitful discussions. MVDE and AK acknowledge support of the Stand Alone Project P33122-N of the Austrian Science Fund (FWF).

---

[3]Recall that a Diophantine equation is a polynomial over the integers whose solutions need to be integers.

AK further acknowledges funding of the Austrian Academy of Sciences (ÖAW) through the DOC scholarship 26547. SS and TR acknowledge support of the START Prize Y 1261-N of the Austrian Science Fund (FWF).

# A    Background on computational complexity

In the following, we summarize the basic notions in computational complexity that are relevant for this paper. For an introduction to the topic, we refer to standard textbooks such as [28, 43].

## A.1    Deterministic and non-deterministic Turing machines

A *(deterministic) Turing machine* is a model of computation consisting of a head, with an internal state, which operates on an infinitely long tape. In words, it works as follows. The input of a Turing machine is initially written on the tape. In each computation step, the head reads off one entry of the tape, it changes its internal state according to the symbol on the tape and its current state, it overwrites the symbol on the tape, and moves one cell left or right. The Turing machine repeats this procedure until it reaches a final state.

More formally, a Turing machine consists of the following: A tape alphabet $\Sigma$ with blank symbol $\sqcup \in \Sigma$, a state set $Q$ with an initial state $q_0$ and final states $F \subseteq Q$, and a transition function

$$\delta : (Q \setminus F) \times \Sigma \to Q \times \Sigma \times \{L, R\}\,,$$

which maps combinations of tape symbol and internal state to a new tape symbol and internal state, together with the instruction to move left or right.

A *non-deterministic Turing machine* is defined similarly, with the only difference that $\delta$ can be a multivalued function, i.e. a tuple $(x, q)$ can map to multiple state-symbol-direction triples. For this reason, a non-deterministic Turing machine has multiple computation paths. A non-deterministic Turing machine halts (within $k$ steps) if there is at least one computation path where it halts (within $k$ steps).

In this work, we often consider Turing machines with empty inputs. This means that every entry on the tape is initially given by the blank symbol $\sqcup$.

## A.2    Decision problems and languages

Decision problems are given by a set of instances together with a question that splits the instance set into yes- and no-instances. A language $L \subseteq \Sigma^*$ of the problem is defined by encoding the set of instances using an alphabet $\Sigma$ and then collecting all yes-instances as elements in $L$. In this work, we address problems and their languages interchangeably via terms like NHALT, PCP, TILE, etc.

## A.3    Complexity classes and reductions

Complexity classes are used to characterize the hardness of decision problems. A language $L$ is *decidable* if there exists a Turing machine $T$ which decides membership in finite time, i.e. if $x \in L$ then $T$ accepts $x$ in finite time, and if $x \notin L$ then $T$ rejects it in finite time.

A language $L$ is *recursively enumerable* (written $L \in$ RE) if and only if there exists a Turing machine $T$ such that for every yes-instance $x \in L$, there exists a finite certificate $y \in \Sigma^*$ for verification, i.e. $\langle x, y \rangle$ is accepted by $T$ in finite time. This means that there exists an algorithm that verifies $x \in L$ in finite time; yet, $x \notin L$ may not be rejected in finite time (since the class decidable is different from RE).

The complexity classes P and NP are defined similarly to decidable and RE, respectively, with the only difference that we ask for efficient (i.e. in polynomial time) solution or verification. Specifically, a problem $L$ is in P if it can be decided by a polynomial-time deterministic Turing machine, i.e. a Turing machine that halts on input $x$ within $p(|x|)$ steps, where $p$ is a fixed polynomial and $|x|$ the length of the input string. A problem $L$ is in NP if and only if there is a polynomial-time Turing machine $T$ such that

$$x \in L \quad \Longleftrightarrow \quad \exists y \in \Sigma^{p(|x|)} : T \text{ accepts } \langle x, y \rangle. \tag{5}$$

A language $L$ is in coNP if and only if its complement $L^c = \Sigma^* \setminus L$ is in NP. Specifically, $L$ is in coNP if and only if there exists a polynomial-time Turing machine $T'$ such that

$$x \in L \quad \Longleftrightarrow \quad \forall y \in \Sigma^{p(|x|)} : T' \text{ accepts } \langle x, y \rangle. \tag{6}$$

Given a polynomial-time Turing machine $T$ verifying $L^c$, $T'$ accepts if and only if $T$ rejects, which proves the statement.

In this work, we are mainly interested in hardness results. For a given complexity class $\mathcal{C}$, a problem is $\mathcal{C}$-hard if it is (in a formal way) at least as hard as any problem in $\mathcal{C}$. The problem is $\mathcal{C}$-complete if it is $\mathcal{C}$-hard and in $\mathcal{C}$.

To formalize these concepts, we need the notion of a reduction between languages. A *reduction from $L'$ to $L$* is a Turing-computable function $\mathcal{R} : \Sigma^* \to \Sigma^*$ which satisfies

$$x \in L' \quad \Longleftrightarrow \quad \mathcal{R}(x) \in L.$$

By abuse of notation, we write $\mathcal{R} : L' \to L$ to highlight the source and target language. If there exists a polynomial-time reduction from $L'$ to $L$, we write $L' \leq_{\text{poly}} L$. Now we can define RE-hardness: A problem $L$ is RE-hard if there exists a reduction $L' \to L$ for every problem $L' \in$ RE. $L$ is RE-complete if $L$ is RE-hard and $L \in$ RE. RE-complete problems are in a formal sense the hardest problems in RE. Similarly, a problem $L$ is NP-hard if for every problem $L' \in$ NP there exists a polynomial-time reduction $\mathcal{R} : L' \to L$. $L$ is NP-complete if it is NP-hard and $L \in$ NP. One defines coNP-hardness analogously.

# B   Complexity of (bounded) halting problems

In this appendix, we provide a detailed analysis of the two halting problems NHALT and NHALTALL together with their bounded versions which act as root problems in the main text (see Section 3). We start with the unbounded problems and their undecidability, and continue with their bounded versions and their complexity.

We start by noting that the input of NHALT and NHALTALL is just a Turing machine $T$, as we ask whether $T$ halts on the empty tape.

**Definition 3.** *Let $T$ be a description of a non-deterministic Turing machine.*

$$T \in \text{NHALT} \quad :\Longleftrightarrow \quad T \text{ halts on the empty tape.}$$

$$T \in \text{NHALTALL} \quad :\Longleftrightarrow \quad \begin{array}{l} T \text{ halts on the empty tape} \\ \text{along all paths.} \end{array}$$

Both problems are undecidable, as the following reduction from the halting problem HALT shows.

**Theorem 4.** NHALT *and* NHALTALL *are* RE-*complete.*

*Proof.* We prove RE-hardness only for NHALT, as the same argument applies to NHALTALL. To this end, we provide a reduction from HALT. Recall that HALT takes $\langle T, x_0 \rangle$ as input (where $T$ is a description of a deterministic Turing machine $T$,[4] and $x_0$ is an input) and accepts if and only if $T$ halts on $x_0$. The reduction transforms instance $\langle T, x_0 \rangle$ to a Turing machine $T' = \mathcal{R}(\langle T, x_0 \rangle)$ which first writes $x_0$ on the tape, and then does the same computation as $T$ on the given input. By construction, $\langle T, x \rangle \in$ HALT if and only if $T' \in$ NHALT, i.e. $\mathcal{R}$ is a valid reduction.

That NHALT $\in$ RE follows by taking the halting computation path as a certificate, and a verifier that verifies the computation along the path. That NHALTALL $\in$ RE follows by taking the halting time as a certificate, and a verifier that verifies that the computation halts along all paths within this halting time. $\qquad\square$

Let us now consider the bounded versions BNHALT and BNHALTALL. Since these problems have different complexity we will treat them separately.

**Definition 5.** *Let $T$ be a description of a non-deterministic Turing machine, and $n \in \mathbb{N}$.*

$$\langle T, 1^n \rangle \in \text{BNHALT} \quad :\Longleftrightarrow \quad \begin{array}{l} \textit{T halts on the empty tape} \\ \textit{in n steps.} \end{array}$$

**Theorem 6.** BNHALT *is* NP-*complete.*

*Proof.* To show that BNHALT is NP-hard, we prove that every NP-language $L$ has a polynomial-time reduction to BNHALT. Since $L$ is in NP, there exists a non-deterministic polynomial-time Turing machine $M$ which accepts $x$ within time $p(|x|)$ if and only if $x \in L$. We construct a non-deterministic Turing machine $P_{M,x}$ that (i) writes $x$ on the tape, (ii) does the same computation as $M$ on the tape with input $x$, and (iii) if $M$ accepts $x$ along a path, $P_{M,x}$ halts along this path, and if $M$ rejects $x$ along a path, $P_{M,x}$ loops along this path. Since step (i) needs a polynomial number $q(|x|)$ steps, and step (iii) needs a constant number $k$ of steps, we have that $x \in L$ if and only if $\langle P_{M,x}, 1^{q(|x|)+k+p(|x|)} \rangle \in$ BNHALT. Completeness follows from Equation (5) by choosing the halting computation path as a certificate, and a polynomial-time verifier which verifies the computation along this path. $\qquad\square$

Similarly, we define the problem BNHALTALL as the language accepting the instance $\langle T, 1^n \rangle$ if and only if $T$ halts on the empty tape along *all* computation paths in at most $n$ steps.

**Definition 7.** *Let $T$ be a description of a non-deterministic Turing machine $T$, and $n \in \mathbb{N}$.*

$$\langle T, 1^n \rangle \in \text{BNHALTALL} \quad :\Longleftrightarrow \quad \textit{T halts on the empty tape along all paths in n steps.}$$

**Theorem 8.** BNHALTALL *is* coNP-*complete.*

*Proof.* The hardness proof is very similar to Theorem 6. Namely, we prove that every coNP-language $L$ has a polynomial-time reduction to BNHALTALL. Since $L$ is in coNP, there exists a non-deterministic polynomial-time Turing machine $M$ which accepts $x$ along every computation path of length at most $p(|x|)$ if and only if $x \in L$. We construct the non-deterministic Turing machine $P_{M,x}$ which (i) writes $x$ on the tape, (ii) does the same computation as $M$ on the tape with input $x$, and (iii) if $M$ accepts $x$ along a path, $P_{M,x}$ halts along this path. If $M$ rejects $x$ along a path, $P_{M,x}$ loops along this path. Since (i) needs a polynomial number $q(|x|)$ steps and (iii) needs a constant number $k$ of steps, we have that $x \in L$ if and only if $\langle P_{M,x}, 1^{q(|x|)+k+p(|x|)} \rangle \in$ BNHALTALL. Completeness again follows from Equation (6) by choosing computation paths as a certificate, and a polynomial-time verifier that verifies the computation along the given path. $\qquad\square$

---

[4]Note that a deterministic Turing machine is a special case of a non-deterministic Turing machine only having one computational path.

# C   More details on undecidable problems and their bounded versions

In this appendix we provide more details on the undecidable problems and their bounded versions considered in the main text. Specifically, we consider the PCP problem (Appendix C.1), the zero in the upper left corner (Appendix C.2), the matrix mortality problem (Appendix C.3) the MPO positivity problem (Appendix C.4), the polynomial positivity problem (Appendix C.5), the stability of positive maps (Appendix C.6), and the tiling problem (Appendix C.7).

## C.1   The PCP problem

We now provide the reduction $\text{NHALT} \to \text{PCP}$ in greater detail. The following reduction modifies that of Ref. [28], so that the bounding parameters of both problems are polynomially related.

We consider a Turing machine given by a tape alphabet $\Sigma$ with blank symbol $\sqcup \in \Sigma$, a state set $Q$ with an initial state $q_0$, final states $F \subseteq Q$, and a transition function

$$\delta : \Sigma \times (Q \setminus F) \to \Sigma \times Q \times \{L, R\}\,.$$

Without loss of generality, we consider here only semi-infinite tape Turing machines, i.e. having a tape with a left end but no right end. This is no restriction for the complexity since semi-infinite tape Turing machines are equivalent to standard Turing machines [43, Claim 1.4].

This Turing machine is mapped to the following set of dominoes $\mathcal{D}$:

(i)   An initial domino
$$\begin{array}{|c|}\hline ! \\ \hline ! \star q_0 \star \sqcup \star ! \star \\ \hline \end{array}$$

(ii)   For every $x \in \Sigma$, a copy domino
$$\begin{array}{|c|}\hline \star x \\ \hline x \star \\ \hline \end{array}$$

(iii)   Transitions $(q, x) \mapsto (\hat{q}, y, L)$
$$\begin{array}{|c|}\hline \star x \star q \\ \hline \hat{q} \star y \star \\ \hline \end{array}$$

(iv)   Transitions $(q, x) \mapsto (\hat{q}, y, R)$
$$\begin{array}{|c|}\hline \star q \star x \\ \hline y \star \hat{q} \star \\ \hline \end{array}$$

(v)   A tape expander
$$\begin{array}{|c|}\hline \star ! \\ \hline \sqcup \star ! \star \\ \hline \end{array}$$

(vi)   For every $q_f \in F$, $y_1, y_2 \in \Sigma$
$$\begin{array}{|c|}\hline \star y_1 \star q_f \star y_2 \\ \hline q_f \star \\ \hline \end{array}$$

(vii)   For every $q_f \in F$, $y_1, y_2 \in \Sigma$
$$\begin{array}{|c|}\hline \star q_f \star y_1 \star y_2 \\ \hline q_f \star \\ \hline \end{array}$$

(viii)   A final domino
$$\begin{array}{|c|}\hline \star q_f \star \sqcup \star ! \star ! \\ \hline ! \\ \hline \end{array}$$

Note that the domino set $\mathcal{D}$ can be constructed in polynomial time from $T$, and that $|\mathcal{D}|$ is polynomial in $|Q|$ and $|\Sigma|$.

Let us now apply this reduction to a non-deterministic Turing machine, as the bounded version needs the latter. First note that the exclamation marks serve as a separator between the instantaneous descriptions of different computation steps, while the grey star separates every symbol in the string. The lower part of the initial domino (i) represents the initial tape configuration of the Turing machine together with its current head state and position. Since the initial domino (i) is the only domino whose first upper and lower symbols coincide, every match has to start with the initial domino. A computation step along some computation path is simulated by applying copy-dominoes (ii), transition dominoes (iii), (iv), and tape expanders (v), according to Figure 4. If a computation reaches a final state $q_f$, the final instantaneous description is successively removed by applying dominoes (ii), (vi), (vii), and (v) according to Figure 4. Finally, a match is obtained by adding (viii).

This implies that $T$ halts on the empty tape along a computation path if and only if $\mathcal{D}$ forms a match. Hence, $\mathcal{R}\colon$ NHALT $\to$ PCP is a reduction. It follows that PCP is RE-hard.

Note that simulating the $k^{\text{th}}$ computation step by a domino arrangement requires precisely $k+1$ dominoes. When $T$ reaches the final state after $n$ computation steps, the post-simulation procedure requires another $n+1$ repetitions, where each procedure needs precisely $m = n+1$ arrangements with length starting with $m$ and decreasing by 1. So $T$ halts after $n$ computation steps on the empty tape if and only if the corresponding domino set forms a match in at most

$$q(n) := 1 + \sum_{k=1}^{n}(k+1) + \sum_{k=1}^{n+1} k = (n+1) \cdot (n+2)$$

steps, where the first sum represents the computation procedure and the second sum the post-simulation procedure. Since $\mathcal{R}$ is a polynomial-time reduction, using Theorem 2, this implies that

$$\langle T, 1^n \rangle \mapsto \langle \mathcal{R}(T), 1^{(n+1)\cdot(n+2)} \rangle$$

is a polynomial-time reduction from BNHALT to BPCP, which shows that BPCP is NP-hard.

## C.2 The Zero in the upper left corner problem

We now present the reduction $\mathcal{R} : $ PCP $\to$ ZULC based on the ideas of [31]. For this purpose, we consider PCP using strings encoded in the alphabet $\Sigma = \{0, 1, 2\}$. We define the bijection $\sigma : \Sigma^* \to \mathbb{N}$ that assigns a representation in base 3 to every natural number, i.e.

$$\sigma(c_1, \ldots, c_n) := \sum_{i=1}^{n} c_i \cdot 3^{n-i}\,.$$

Moreover, we define a function $\gamma : \Sigma^* \times \Sigma^* \to \mathbb{N}^{3\times 3}$ via

$$\gamma(w_1, w_2) := \begin{pmatrix} 3^{|w_1|} & 0 & 0 \\ 0 & 3^{|w_2|} & 0 \\ \sigma(w_1) & \sigma(w_2) & 1 \end{pmatrix}.$$

The function $\gamma$ is injective and a morphism, i.e. $\gamma(w_1 u_1, w_2 u_2) = \gamma(w_1, w_2) \cdot \gamma(u_1, u_2)$ where composition on $\Sigma^*$ is given by concatenation of words. Let

$$d_1 = \begin{bmatrix} a_1 \\ b_1 \end{bmatrix}, \ldots, d_k = \begin{bmatrix} a_k \\ b_k \end{bmatrix}$$

be an instance of PCP where $a_i, b_i \in \Sigma^*$. For $i \in \{1, \ldots, k\}$, we define the matrices

$$A_i = X \cdot \gamma(a_i, b_i) \cdot X^{-1} \quad B_i = X \cdot \gamma(a_i, 0 b_i) \cdot X^{-1},$$

with

$$X = \begin{pmatrix} 1 & 0 & 1 \\ 1 & 1 & 0 \\ 0 & 0 & 1 \end{pmatrix}.$$

We have that

$$d_{i_1} d_{i_2} \cdots d_{i_n}$$

is a matching domino if and only if

$$(M_{i_1} \cdot M_{i_2} \cdots M_{i_n})_{11} = 0,$$

where $M_{i_j} \in \{A_{i_j}, B_{i_j}\}$. We refer to [31] for details. This shows that $\mathcal{R} : \text{PCP} \to \text{ZULC}$ with

$$\mathcal{R}\big(\langle d_1, \ldots, d_k \rangle\big) := \langle A_1, \ldots, A_k, B_1, \ldots, B_k \rangle$$

is a polynomial-time reduction. This implies that ZULC is RE-hard.

Since matches of length $n$ are mapped to matrix multiplications of length $n$ with a zero in the upper left corner, this shows that $\mathcal{R}_b : \text{BPCP} \to \text{BZULC}$ with

$$\mathcal{R}_b\big(\langle d_1, \ldots, d_k, 1^n \rangle\big) := \langle A_1, \ldots, A_k, B_1, \ldots, B_k, 1^n \rangle$$

is a polynomial-time reduction. This implies that BZULC is NP-hard.

Note that the matrices in $A_1, \ldots, A_k, B_1, \ldots, B_k$ are invertible, from which it follows that ZULC and BZULC remain RE-hard and NP-hard, respectively, when restricting the instances to invertible matrices.

## C.3 The Matrix mortality problem

We now construct the reduction $\mathcal{Q} : \text{ZULC} \to \text{MM}$ following the ideas of [31]. Since ZULC remains hard when restricting the instances to invertible matrices, we construct $\mathcal{Q}$ only for invertible matrices. So let $\langle A_1, \ldots, A_k \rangle$ be an instance of invertible matrices in ZULC. We define

$$\mathcal{Q}(\langle A_1, \ldots, A_k \rangle) := \langle A_1, \ldots, A_k, B \rangle,$$

with

$$B = \begin{pmatrix} 1 & 0 & 0 \\ 0 & 0 & 0 \\ 0 & 0 & 0 \end{pmatrix}.$$

We claim that $A_1, \ldots, A_k$ forms a zero in the upper left corner if and only if $A_1, \ldots, A_k, B$ multiplies to a zero matrix. This proves that MM is RE-hard. Moreover, we show that

$$n_{\min,\text{MM}}[\langle \mathbf{A}, B \rangle] = n_{\min,\text{ZULC}}[\langle \mathbf{A} \rangle] + 2, \tag{7}$$

where $\mathbf{A}$ represents the list $A_1, \ldots, A_k$.

To prove the claim, first note that if

$$(A_{i_1} \cdot A_{i_2} \cdots A_{i_n})_{11} = 0,$$

then

$$B \cdot A_{i_1} \cdot A_{i_2} \cdots A_{i_n} \cdot B = (A_{i_1} \cdot A_{i_2} \cdots A_{i_n})_{11} = 0.$$

In other words, a yes-instance of ZULC with parameter $n$ is mapped to a yes-instance in MM with parameter $n + 2$. This proves the inequality "≤" of Equation (7).

Conversely, assume that there exists a sequence of $n$ matrices in $\{A_1, \ldots A_k, B\}$ that multiplies to $\mathbf{0}$. Since $A_1, \ldots, A_k$ are invertible and $B$ has rank 1, this sequence must contain $B$ at least twice. The product is of the form

$$M_1 B M_2 B M_3 B \cdots B M_r = \mathbf{0},$$

where $M_i$ is a multiplication of $\ell_i$ matrices in $\{A_1, \ldots, A_k\}$ for some $\ell_i$.[5] Since $B$ is idempotent, we have that

$$
\begin{aligned}
0 &= \left(M_1 B M_2 B M_3 B \cdots B M_r\right)_{11} \\
&= \left(B M_1 B^2 M_2 B^2 M_3 B^2 \cdots B^2 M_r B\right)_{11} \\
&= \left(M_1\right)_{11} \cdots \left(M_r\right)_{11}.
\end{aligned}
$$

This implies that at least one of the matrices $M_i$ has a zero in the upper left corner, which shows that $A_1, \ldots, A_k$ form a zero in the upper left corner with a word of length $n$. Specifically, any minimal sequence of matrices realizing $\mathbf{0}$ must be of the form

$$B \cdot A_{i_1} \cdot A_{i_2} \cdots A_{i_n} \cdot B = \mathbf{0}.$$

Note that a shorter such product cannot exist because it would violate the proven inequality "≤" of Equation (7). This representation proves the inequality "≥" of Equation (7), since

$$(A_{i_1} \cdot A_{i_2} \cdots A_{i_n})_{11} = 0.$$

In summary, $\mathcal{Q}\colon \text{ZULC} \to \text{MM}$ is a reduction, which proves that MM is RE-hard. Moreover, $\mathcal{Q}_b\colon \text{BZULC} \to \text{BMM}$ with

$$\mathcal{Q}_b \colon \langle A_1, \ldots, A_k, 1^n\rangle \mapsto \langle A_1, \ldots, A_k, B, 1^{n+2}\rangle$$

is a polynomial-time reduction too, which proves that BMM is NP-hard.

## C.4 The MPO positivity problem

Here we present a reduction $\mathcal{R}\colon \text{ZULC} \to \text{MPO}$, slightly different than [8]. The MPO problem has as input a fixed number of $D \times D$ integer matrices $\langle B_i : i \in \{1, \ldots, k\}\rangle$ and asks whether there exists a natural number $n \in \mathbb{N}$ such that

$$\rho_n(B) := \sum_{i_1, \ldots, i_n = 1}^{k} \operatorname{tr}\left(B_{i_1} \cdots B_{i_n}\right) |i_1 \ldots i_n\rangle \langle i_1 \ldots i_n|$$

is not positive semidefinite. We define

$$\mathcal{R}(\langle A_1, \ldots, A_k\rangle) = \langle B_1, \ldots, B_k, B_{k+1}\rangle,$$

where for $i \in \{1, \ldots, k\}$

$$B_i := \begin{pmatrix} A_i \otimes A_i & 0 \\ 0 & 1 \end{pmatrix},$$

and

$$B_{k+1} := \begin{pmatrix} E_{11} & 0 \\ 0 & -1 \end{pmatrix},$$

---

[5]If it is an empty multiplication (i.e. $\ell_i = 0$), then we define $M_i$ as the identity matrix.

where $E_{11} := \mathbf{e}_1 \mathbf{e}_1^t$ with $\mathbf{e}_1 = (1, 0, \ldots, 0)^t$ of length $D$.

We now prove that the threshold parameter $n$ in BZULC maps to the threshold parameter $n + 1$ in BMPO. Let $A_{i_1}, \ldots, A_{i_n}$ be the minimal sequence such that

$$\left( A_{i_1} \cdot A_{i_2} \cdots A_{i_n} \right)_{11} = 0 \, .$$

Then,

$$\mathrm{tr}(B_{i_1} \cdots B_{i_n} \cdot B_{k+1}) = \left( A_{i_1} \cdot A_{i_2} \cdots A_{i_n} \right)_{11}^2 - 1 < 0 \, .$$

Conversely, let $B_{i_1}, \ldots, B_{i_{n+1}}$ be a minimal sequence such that

$$\mathrm{tr}(B_{i_1} \cdot B_{i_2} \cdots B_{i_{n+1}}) < 0 \, .$$

The indices $i_1, \ldots, i_{n+1}$ cannot be chosen exclusively from $\{1, \ldots, k\}$, since in that case

$$\mathrm{tr}(B_{i_1} \cdot B_{i_2} \cdots B_{i_{n+1}}) = \left( \mathrm{tr}(A_{i_1} \cdots A_{i_{n+1}}) \right)^2 + 1 \geq 0 \, .$$

Hence, there is at least one index $i_\ell = k + 1$. Assume that there is precisely one index $k + 1$. Without loss of generality, we assume $i_{n+1} = k + 1$ due to cyclicity of the trace. This leads to

$$0 > \mathrm{tr}(B_{i_1} \cdot B_{i_2} \cdots B_{i_{n+1}}) = \left( \left( A_{i_1} \cdot A_{i_2} \cdots A_{i_n} \right)_{11} \right)^2 - 1 \, ,$$

which implies that $\left( A_{i_1} \cdot A_{i_2} \cdots A_{i_n} \right)_{11} = 0$ because the entries are integer. This shows that a threshold parameter $n + 1$ in BMPO maps to a threshold parameter of a most $n$ in BZULC. Note that having multiple indices with $k + 1$ leads to a smaller threshold parameter in BZULC which contradicts the minimality assumption of $B_{i_1}, \ldots, B_{i_{n+1}}$. This proves the statement.

This reduction can easily be extended to matrices with rational numbers.

In summary, $\mathcal{R} \colon$ ZULC $\to$ MPO is a reduction, which proves that MPO is RE-hard. Moreover, by Theorem 2, $\mathcal{R}_b \colon$ BZULC $\to$ BMPO with

$$\mathcal{R}_b : \langle A_1, \ldots, A_k, 1^n \rangle \mapsto \langle B_1, \ldots, B_k, B_{k+1}, 1^{n+1} \rangle$$

is a polynomial-time reduction too, which proves that BZULC is NP-hard.

### C.5 The Polynomial positivity problem

Let us now review the reduction $\mathcal{R} :$ MPO $\to$ POLY from [13]. We define

$$\mathcal{R}(\langle B_1, \ldots, B_k \rangle) := \left\langle q_{\alpha, \beta}(\mathbf{x}) : \alpha, \beta = 1, \ldots, D \right\rangle \, ,$$

with

$$q_{\alpha, \beta}(\underline{x}) := \sum_{j=1}^{k} \left( B_j \right)_{\alpha, \beta} x_j^2 \, ,$$

where $\underline{x}$ is a $k$-tuple of variables. It is clear that $\mathcal{R}$ is a polynomial-time function. We now prove that $\mathcal{R}$ is a reduction.

If there exists a sequence of matrices such that

$$\mathrm{tr}(B_{i_1} \cdot B_{i_2} \cdots B_{i_n}) < 0 \, ,$$

then

$$p_n(\mathbf{e}_{i_1}, \ldots, \mathbf{e}_{i_n}) = \mathrm{tr}(B_{i_1} \cdot B_{i_2} \cdots B_{i_n}) < 0 \, ,$$

where $p_n$ is defined in (3) and $\mathbf{e}_\ell$ is the $\ell$th standard vector. This implies that $p_n$ is *not* a nonnegative function. Conversely, if

$$\mathrm{tr}(B_{i_1} \cdot B_{i_2} \cdots B_{i_n}) \geq 0 \, ,$$

for all indices $i_1, \ldots, i_n$, then $p_n$ is a sum-of squares which is also nonnegative.

This proves that the threshold $n$ for BMPO is mapped to the threshold $n$ for BPOLY. It follows that BPOLY is NP-hard. Moreover, POLY is RE-complete and BPOLY is NP-complete by taking an arrangement of the matrices leading to a negative value as a certificate, and a polynomial-time verification procedure of this statement as a verifier.

## C.6   Stability of positive maps

Let us now review the reduction $\mathcal{R} : \text{MPO} \to \text{TSP}$ of [14], which proves that TSP is RE-hard. The same reduction also yields that BTSP is NP-hard.

We map an instance

$$\langle B_1, \ldots, B_k \rangle \in \mathcal{M}_{D^2}(\mathbb{Q}) \cong \mathcal{M}_D(\mathbb{Q}) \otimes \mathcal{M}_D(\mathbb{Q})$$

of MPO to a linear map

$$\mathcal{P} : \quad \mathcal{M}_D(\mathbb{Q}) \otimes \mathcal{M}_D(\mathbb{Q}) \quad \to \quad \mathcal{M}_k(\mathbb{Q}),$$
$$X \quad \mapsto \quad \sum_{i=1}^{k} |i\rangle \langle i| \, \text{tr}(C_i X),$$

where

$$(C_i)_{(\alpha_1, \alpha_2),(\beta_1, \beta_2)} := (B_i)_{(\alpha_1, \beta_1),(\alpha_2, \beta_2)},$$

with $\alpha_1, \alpha_2, \beta_1, \beta_2 \in \{1, \ldots, D\}$. Then, we have that

$$\text{tr}\left(C_{i_1} \otimes \cdots \otimes C_{i_n} \chi_n\right) = \text{tr}(B_{i_1} \cdots B_{i_n}),$$

where $\chi_n$ is defined in (4). By construction, this implies that

$$\mathcal{P}^{\otimes n}(\chi_n) = \rho_n(B).$$

In summary, $\langle B_1, \ldots, B_k \rangle \in \text{MPO}$ if and only if exists $n \in \mathbb{N}$ such that $\mathcal{P}^{\otimes n}(\chi_n) \not\geq 0$. Further the threshold parameters in both problems coincide for this reduction. It follows that BTSP is NP-hard.

## C.7   The tiling problem

We now review the reduction $\mathcal{R} : \text{HALT} \to \text{TILE}$ from [37]. A Turing machine, consisting of a tape alphabet $\Sigma$ with blank symbol $\sqcup \in \Sigma$, a state set $Q$ with an initial state $q_0$ and final states $F \subseteq Q$, and a transition function

$$\delta : \Sigma \times (Q \setminus F) \to \Sigma \times Q \times \{L, R\}$$

is mapped to the following set of tiles:

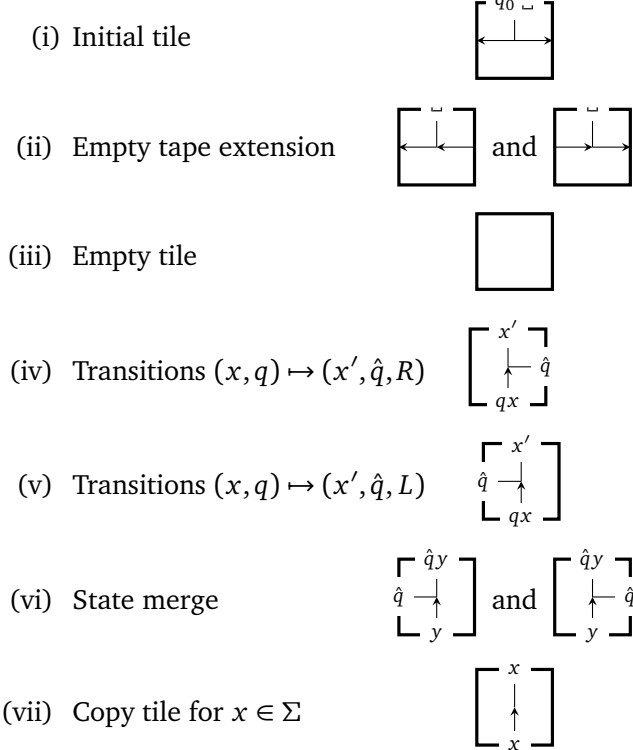

(i) Initial tile

(ii) Empty tape extension and

(iii) Empty tile

(iv) Transitions $(x,q) \mapsto (x',\hat{q},R)$

(v) Transitions $(x,q) \mapsto (x',\hat{q},L)$

(vi) State merge and

(vii) Copy tile for $x \in \Sigma$

Note that (vi), State merge, is defined for every $y \in \Sigma$ and $\hat{q} \in Q$, whereas (iv) and (v), Transitions, are defined for every such tuple in $\delta$.

This set of tiles captures the computation of a Turing machine on the empty tape when placing the initial tile to the origin (see Figure 7). The initial tile can only be extended to the left and to the right with (ii) Empty tape extensions. We can also trivially tile the whole lower half of the plane by applying the empty tile. The generated string

$$\cdots \quad \sqcup \quad \sqcup \quad \sqcup \quad q_0 \sqcup \quad \sqcup \quad \sqcup \quad \sqcup \quad \cdots$$

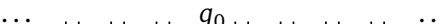

at the top of the first line represents the instantaneous description of the Turing machine at time 0, namely an empty tape with the head at position 0 and state $q_0$. Simulating one step of the Turing machine corresponds to filling up the line above of the current one. Specifically, on the top of the initial tile, we need to place a Transition tile ((iv) or (v)) $(q_0, \sqcup) \mapsto (\hat{q}, x, L/R)$. Then we need to place a (vi) State merge tile on the left/right of the transition tile. This reflects the movement of the head to the left or right. The rest of the line is filled with (vii) Copy tiles.

Again, the string at the top of the second line represents the initial description after one computation step. The same procedure applies to every computation step. As soon as we apply a transition tile $(q, x) \mapsto (q_f, y, L/R)$ for some final state $q_f \in F$, there is no tile to continue the tiling procedure. In other words, every tiling procedure terminates in line $n$ if and only if $T$ halts on the empty tape.

The same reduction applies to non-deterministic Turing machines. In this situation, every tiling procedure terminates in $n$ lines if and only if the Turing machine halts on the empty tape along every computation path in at most $n$ steps. In other words, a Turing machine $T$ halts on every path in at most $n$ steps if and only if $\mathbb{Z}_{n+1} \times \mathbb{Z}_{n+1}$ cannot be tiled. This proves that $\mathcal{R} : \text{NHALTALL} \to \text{TILE}$ is a reduction. It follows that TILE is RE-hard.

Moreover, $\mathcal{R}$ is a polynomial-time map. Since the map between the threshold parameters of NHALTALL and TILE is given by $n \mapsto n + 1$,

$$\langle x, 1^n \rangle \mapsto \langle \mathcal{R}(x), 1^{n+1} \rangle$$

is a reduction from BNHᴀʟᴛAʟʟ to BTɪʟᴇ. This implies that BTɪʟᴇ is coNP-hard.

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
