# Peer review of "Many bounded versions of undecidable problems are NP-hard"

_SciPost Physics, doi:SciPost Phys. 14, 173 (2023)_

## Round 2 · Referee Report · Anonymous · 2023-1-8

Strengths

1. The paper is very well written, I think it contains sufficient background on computational complexity for readers with a physics background to understand the results and their significance.

2. The authors shed light on the relationship between bounded and unbounded problems, and in particular on the relationship between undecidability and hardness. This is an interesting area which to the best of my knowledge has not been rigorously explored before.

Weaknesses

1. The paper in its entirety contains a very good overview of computational complexity for the non expert, and thorough descriptions of all the computational problems used. However sometimes computational problems / computational complexity terms are referenced prior to their definition. This may put off some readers (a possible solution would be to update Appendix A to also include descriptions of all computational problems used in the manuscript, and reference it in the introduction for any readers who aren't familiar with complexity theory).

2. There isn't much discussion of why the distinction between undecidability and hardness is interesting from a physical point of view - I think this would be nice to include (particularly since the submission is being considered for SciPost Physics).

Report

This paper tackles the question of whether the observation that many undecidable problems have NP-hard bounded versions can be rigorously understood. The authors derive a method that leverages the standard techniques for proving undecidability in order to derive hardness results for bounded versions of certain types of undecidable problems.

The paper begins by explaining what a bound version of a problem is, and deriving the relationship between reductions between unbound problems and reductions between bound problems. It then goes on to introduce two bound versions of the halting problem, and finally it uses reductions from the halting problem to a number of other problems to demonstrate NP / co-NP hardness of a number of bound problems that are known to be undecidable in the unbounded case.

The question that is tackled by these authors is an extremely interesting one at the intersection of physics and computational complexity. I believe that the approach taken by the authors to tackling it is elegant, and the results in the paper are well motivated and insightful. As detailed in section 5 there are a number of other research questions which could now be tackled using this paper as a starting point. I believe this paper meets the criteria for publication in SciPost Physics.

Given the very short turnaround time for the review I have not been able to check all the proofs rigorously, but to the best of my understanding the proofs in the paper are correct.

Requested changes

1. In theorem 2 it's not completely clear to me why p has to be strictly increasing as opposed to just non-decreasing. Could the authors clarify this point.

2. In the proof of theorem 4 it might be helpful to note in a footnote that a deterministic Turing machine is a special case of a non-deterministic Turing machine to help explain the reduction from Halt to NHalt.

3. Including a bit more detail about the complexity class co-NP in appendix A could help readers when they come to the proof of theorem 8 (in particular including a sentence about why the definition of co-NP implies acceptance on all computational paths).

4. In section 4A the abbreviation MPO is used for the first time but not explained until section 4C.

5. Page 4, bottom of column 1, where it says "it is to be NP-complete" I think it should be "it is known to be NP-complete"

6. Page 9, bottom of column one, where it says "of the latter use the dimension" should this be "of the latter uses the dimension"

7. In section 5 could the others include some discussion of whether there is any hope of understanding an implication in the opposite direction (maybe there are obvious counterexamples showing not - if so could the authors reference them).

8&9. The points from the weaknesses section could also be tackled.

---

## Round 2 · Referee Report · Anonymous · 2023-2-14

Strengths

1-Very good overview for some NP-hard problems and their corresponding undecidable formulations , which were analyzed in (quantum) information science literature.
2-The authors show a rigorous relationship between the undecidable problems and their "bounded" versions.
3-Generally well written with good figures and a good, succinct introduction for the considered problems.

Weaknesses

1- The overall ordering of the paper is sometimes confusing. Concepts are discussed, but defined only later.
2- The authors could spend more time explaining how their main result, theorem 2, fits within the broader complexity theory literature.

Report

The paper is well written and I could not spot any technical errors. For physicists wanting to show complexity theoretic hardness for their respective problems, this paper provides a good summary of different proof techniques.

The main theoretical contribution is showing that a reduction using NHalt (NHaltAll) or PCP also gives NP (coNP)-hardness for the bounded version when the requirements of theorem 2 are met. While this is not a surprising result for complexity theorists, it might proof to be a useful tool in future hardness proofs.

Overall , I think if the authors perform some minor changes, the work should be published SciPost Physics.

Requested changes

-It is not always clear which aspects and ideas are the authors own contribution, and which originate from literature. In particular, have there already been studies on "bounded" languages (as in def. 1) in literature?

-In physics, the approximate version of problems (eg. $\epsilon$-closeness to positive MPOs in some norm) might be more relevant than the undecidability for infinite system size. Could the authors comment, if their work can also be used to show approximate NP-hardness in the finite case?

Small comments:

2.A:
-x refers to two different concepts: an element in a language and an initial state of a Turing machine. Maybe use different symbols?
- BHalt seems to be the natural bounded version of Halt. (which is P-complete). I think it would improve readibility, if the authors made clear that NHalt is used because it is essential for later reductions, as BHalt would not work.
2.B:
Eq. 1: does n need to be an exact polynomial, or just upper-bounded by one?

3.:
The final paragraph can only be understood after having read the paper. Even when one is familiar with the problems, the (coNP) formulation of the tiling problem was not mentioned thus far, making the NHalt vs NHaltAll discussion hard to follow.

4A:
For PCP, n is the number of words (effectively number of dominos)..In BPCP it is the maximal size of the matching. Maybe use k as the number of input words (dominos) for both?

Spelling: "It is to be NP-complete" -> "Is known to be NP-complete"

4C:
Why are the MPO are chosen to describe diagonal operators? This does not seem to be the canonical definition.

4G:
Could the authors elaborate, why NHaltAll needed to be used for Wang tiling problems in particular?

---

## Round 3 · Author Response

We thank the referees for their work and constructive comments on our manuscript.

---

## Round 3 · List of Changes

1. Following the suggestion of both referees, we clarified in Section 2 that there is, to the best of our knowledge, no prior work on bounding from a systematic perspective.
2. Following the suggestion in Report 1, we added a discussion about whether a converse statement to the main theorem also holds. This includes a comment on unbounding in Section 2 and a new paragraph in Section 5.
3. Following Report 1, we clarified why p needs to be strictly increasing in Theorem 2. Moreover, we weakened the assumption in Equation (1).
4. We restructured the presentation of the examples HALT, NHALT, and NHALTAll. Specifically, we removed NHALT as an example of bounding in Section 2 and directly introduced it as a root problem in Section 3. Moreover, we added a paragraph in Section 3 that clarifies that the deterministic Halting problem HALT is too weak to act as a root problem.
5. We changed the caption in Figure 2 to explain every abbreviation in the figure.
6. Following Report 2, we added a paragraph in Section 4.C clarifying the use of diagonal MPOs instead of the general definition.
7. Following Report 2, we added the approximate MPO problem and its bounded version as an example highlighting that our theorem can be used for problems containing approximation errors.
8. Following Report 2, we added a paragraph in Section 4.G. to explain why it is necessary to use NHALTAll instead of NHALT as a root problem for the Wang tiling problem.
9. Following Report 1, we elaborate on the definition of coNP in Appendix A.c and clarify its use when proving the completeness of BNHALTAll.

You are currently on this page

Resubmission 2211.13532v3 on 16 March 2023

---

## Editorial Decision

published